# Immunomodulatory Effects of Radiotherapy

**DOI:** 10.3390/ijms21218151

**Published:** 2020-10-31

**Authors:** Sharda Kumari, Shibani Mukherjee, Debapriya Sinha, Salim Abdisalaam, Sunil Krishnan, Aroumougame Asaithamby

**Affiliations:** 1Division of Molecular Radiation Biology, Department of Radiation Oncology, University of Texas Southwestern Medical Center, Dallas, TX 75390, USA; Sharda.Kumari@utsouthwestern.edu (S.K.); Debapriya.Sinha@utsouthwestern.edu (D.S.); Salim.Abdisalaam@utsouthwestern.edu (S.A.); 2Department of Radiation Oncology, Mayo Clinic Florida, Jacksonville, FL 32224, USA; Krishnan.Sunil@mayo.edu

**Keywords:** radiation therapy, charged particle therapy, carbon ion therapy, clustered DNA damage, immune signaling, cancer vaccines, tumor antigens, abscopal effects, FLASH-RT

## Abstract

Radiation therapy (RT), an integral component of curative treatment for many malignancies, can be administered via an increasing array of techniques. In this review, we summarize the properties and application of different types of RT, specifically, conventional therapy with x-rays, stereotactic body RT, and proton and carbon particle therapies. We highlight how low-linear energy transfer (LET) radiation induces simple DNA lesions that are efficiently repaired by cells, whereas high-LET radiation causes complex DNA lesions that are difficult to repair and that ultimately enhance cancer cell killing. Additionally, we discuss the immunogenicity of radiation-induced tumor death, elucidate the molecular mechanisms by which radiation mounts innate and adaptive immune responses and explore strategies by which we can increase the efficacy of these mechanisms. Understanding the mechanisms by which RT modulates immune signaling and the key players involved in modulating the RT-mediated immune response will help to improve therapeutic efficacy and to identify novel immunomodulatory drugs that will benefit cancer patients undergoing targeted RT.

## 1. Introduction

Cancer is a disease of uncontrolled cellular proliferation, leading to unfaithful and uncoordinated DNA replication, genomic instability, and DNA double-strand breaks (DSBs). Though the principle of using DNA damage to kill tumor cells has been applied for decades, the problem of how to kill the tumor but not the normal cells has remained the same. Furthermore, cancer cells often exhibit defects in the DNA repair mechanism that consequently drive genomic instability, thereby fueling further tumorigenesis. Paradoxically, these defects in DNA repair can be exploited by inhibitors that target the remaining DNA repair pathways that the cancer cells rely upon more heavily, not only because of the loss of other repair factors but also because of the higher load of overall DNA damage in cancer cells.

Clinicians use chemotherapy, radiation therapy, and immunomodulatory agents to treat various cancers with the hope of achieving higher response rates and improving the overall survival rate. Of the most current anti-cancer therapies, particle therapy (PT) is emerging as a promising modality for cancer treatment worldwide [1]. PT beam dose deposition follows the Bragg curve as a function of depth in tissue. The lower entrance doses and the elimination of exit doses can produce superior conformal dose distributions on the target volume as compared with conventional radiotherapy (RT) with x-rays. Although PT clinical practice uses proton or Carbon-ion (12C) beams, other light ion beams, such as Helium, Oxygen, and Neon, have also been investigated [2,3,4]. These particles each have their own physical and biological advantages. 12C beams are attractive, because they provide sharp lateral penumbra, sharp distal dose fall off, less dependence on hypoxic conditions, less dependence on cell cycle phase, and higher relative biological effectiveness (RBE), which may be advantageous for treating radioresistant tumors that require superior dose conformity [5,6]. Thus, the rationale for using PT beams to successfully treat cancer patients lies in reducing the integral dose and sparing surrounding healthy tissues and critical organs, minimizing treatment-related complications, and reducing the risk of radiation-induced secondary cancers [7,8].

Using RT in conjunction with immunomodulatory agents can trigger an antitumor immune response within the body that leads to a higher treatment response rate and improved patient survival, but these cytotoxic therapies may also destroy immune and hematopoietic stem cells and even stimulate regrowth in residual malignant cells. To fully exploit the clinical potential of RT-induced antitumor immunity while avoiding immunosuppression, we need to understand not only the radiobiology but also the underlying mechanisms of immunomodulatory effects of RT.

## 2. Radiation as Therapy

Radiation therapy (RT) is one of the primary modalities for treating different types of cancer, either as a standalone therapy or in combination with chemotherapy or immunotherapy administered as a definitive therapy or in conjunction with surgery. RT’s main purpose is to eliminate tumor cells while sparing the nearby normal tissues from radiation damage. In multimodal therapy, RT is given either before (neoadjuvant), during (concurrent), or after (adjuvant) another treatment modality. RT is delivered to the tumor either by external beams focused on the tumor or by directly embedding radioactive sources into the tumor site (brachytherapy). Depending on the volume, location, and sensitivity of the tumor and the surrounding normal tissue, different types of RT—such as x-rays protons, and carbon ions—may be selected.

### 2.1. Conventional Photon Radiation Therapy

Treatment with ionizing radiation has been, and will continue to be, an essential modality of cancer therapy, as more than 60% of all cancer patients receive conventional radiation (γ- or X-rays) at some point during the course of their treatment. Historically, the total prescribed dose has been divided into many smaller fractions to spare substantial amounts of normal tissue within the treatment field. Hence, RT has to balance delivering a sufficient dose to eradicate the tumor against limiting the real risk for adverse responses in normal tissue. Technological advances such as intensity-modulated and image-guided RT have improved the delivery of the therapeutic dose specifically to the tumor volume, thus increasing the differential between tumor control probability and normal tissue complications. Other advances, such as stereotactic body radiotherapy (SBRT), have enhanced dose conformity to substantially limit treatment margins, so ablative doses of RT can be delivered to the tumor in five or fewer fractions.

### 2.2. Stereotactic Body Radiotherapy (SBRT)

Conventional RT is effective when it is combined with surgery. When tumors present in areas where surgery is not possible, long-term survival with conventional RT is lower. SBRT—which uses numerous small, highly focused, accurate radiation beams to deliver potent doses in one to five treatments—can ablate tumors in surgically inoperable locations while sparing the sensitive normal tissues around the tumor [9]. SBRT achieves greater than 90% local control and improves survival over conventional RT with minimal toxicity in peripherally located stage I non-small cell lung cancer (NSCLC) [10,11]. Patients with inoperable NSCLC who received SBRT had better 3-year overall survival with better local tumor control than patients who were treated with conventional RT [10]. SBRT in older adults with early stage lung cancer showed the same trend [12,13]. In some studies, patients treated with SBRT for bone metastases in oligometastatic prostate carcinoma [14], hepatocellular carcinoma [15], prostate cancer [16,17], head and neck cancer [18], and pancreatic cancer [19] also showed better outcomes than patients who received conventional RT. Nonetheless, SBRT’s higher dose per fraction requires that the effective dose essentially hit only the tumor to minimize radiation-related damage to surrounding normal tissues. A certain amount of damage to these tissues is unavoidable, however, as the radiation must penetrate to the target from many directions. As a result, normal tissues adjacent to the cancer—for example, esophagus, bronchial tree, and major vessels near lung tumors—ultimately limit the maximum SBRT dose [20,21].

### 2.3. Charged Particle Therapy

Charged particle therapy uses protons or heavier charged particles, instead of γ- or X-rays, to treat cancer. One of the major advantages of proton and heavier charged particles is their favorable depth dose distributions compared to those of the electrons and photons used in conventional RT [22,23]. When an accelerated charged particle moves through matter, it ionizes the material’s atoms and deposits a dose along its path. An energy deposition peak occurs as the interaction between the charged particles and the cross-section of tissues increases. Gradually, the energy of the charged particle decreases. Energy lost by charged particles is inversely proportional to the square of their velocity, so the energy peaks just before the particle comes to a complete stop. This peak is called the Bragg peak, named after William Henry Bragg, who discovered it. Particle therapy treatment for cancer exploits the Bragg peak phenomenon to concentrate the effect of ion beams on the tumor while minimizing effects on the surrounding healthy tissues. Moreover, the exit dose—that is, the radiation exposure past the target, which can damage nearby organs and normal tissues and cause future health issues—is exceedingly small in charged particle therapy. Ultimately, charged particles’ energy deposition properties allow particle therapy to achieve greater dose conformality than other RT modalities.

#### 2.3.1. Proton Therapy

Using protons in medical treatment was first proposed in 1946 [24]. There are now more than 30 proton therapy centers in the U.S. alone. Proton therapy is a better choice than conventional RT because it deposits a relatively constant dose along the beam path until the end of its range, where the dose peaks and then falls off to near-zero quite rapidly [25]. Therefore, this therapy can minimize the radiation dose to surrounding normal tissues, which may be sensitive to radiation. One of the drawbacks of proton therapy is that its initial cost of treatment is much higher than that of conventional RT. Nonetheless, proton therapy’s overall total cost is substantially lower than conventional therapy’s because there are fewer adverse responses in normal tissues that can drive up total treatment costs [22,23].

More and more, proton therapy is becoming a better option for treating many types of cancer located near sensitive normal tissue (Table 1). Proton therapy is useful for sparing the normal tissues near tumors, but lateral scattering still limits its biological benefits over conventional RT. Due to this, heavier ions, like carbon ions, which have less lateral scattering but still deposit energy at the tumor, are increasingly part of the discussion of treatment options for different types of cancers [26].

#### 2.3.2. Carbon Ion Therapy

^12^C ions are the optimal ion for treating deep-seated tumors, because the higher relative biological effectiveness (RBE) resulting from variations in linear energy transfer (LET) along the ion path can be limited to the tumor volume with minimal normal tissue injury along the entrance track. In addition to their favorable RBE characteristics, ^12^C ions offer minimal lateral and longitudinal scattering—one-third that of protons—and the potential to use in-line positron emission tomography (PET) for dose deposition verification and mapping. Furthermore, ^12^C ions’ higher RBE will probably make them more efficacious against radioresistant tumors, and hypoxic tumors will probably also respond better to high-LET radiation than to megavoltage RT or protons. Hepatocellular carcinoma, lung cancer, and head and neck cancer are some of the tumors that could benefit from ^12^C ion therapy (CIRT) [52,53,54]. In addition, ^12^C ions’ energy deposition patterns result in more complex DNA lesions that are more difficult to repair, so combined treatment with DNA repair inhibitors may hold greater promise.

National Institute of Radiological Sciences (NIRS) in Chiba, Japan leads in treating patients with carbon ions and it has treated more than 13,000 patients since 1994 [55]. Relying on their experience with fast neutrons, the NIRS team determined the RBE for their passive system across a spread-out Bragg Peak by using either Chinese hamster ovary cells or, ultimately, human salivary gland (HSG) cells in their microdosimetric kinetic model (MKM) [56]. This model could be modified to integrate cell sensitivity, fraction number and dose-dependent RBE. Particle therapy centers in Europe, on the other hand, use the local effects model (LEM), which has been modified to include track structure (LEM III) and now biological effects (LEM IV) [57]. Both the MKM and LEM models can be used for both tumor and normal tissue responses, and as the biological data grow, so do the accuracy and the specificity to tumor type or normal tissue endpoints. Incorporating the hypoxia response and combinatorial agents into these models would also be appropriate. Although these models have been implemented in clinical treatment planning systems, other biological endpoints, such as hypoxia or intrinsic radiosensitivity, have not yet been integrated.

Even though 13,000 individuals have been treated with ^12^C ions, no randomized phase III clinical trials have been completed to date. Nonetheless, an abundance of evidence from phase II studies suggests that CIRT achieves excellent treatment responses with favorable safety profiles, even though treatment regimens were developed empirically [55]. For instance, treating skull base chordomas and chordosarcomas with CIRT achieves excellent tumor control and overall survival with a low level of toxicity [58,59,60]. The possibility that gastrointestinal cancers may benefit from CIRT also looks promising [61]. Patients with prostate cancer treated with CIRT showed better outcomes than those treated with conventional RT [62,63]. It is increasingly assumed that accelerated carbon ions may be a better treatment option than conventional RT for deep-seated, radioresistant, and hypoxic tumors [60]. However, for charged particle therapy to succeed in the future, it must have a foundation based upon evidence derived from basic, preclinical, and translational research. As with conventional therapy, initial improvements in charged particle therapy were driven predominantly by advances in physics, engineering, and imaging; however, future advances with charged particles, including protons, will need to be driven predominantly by biological science.

## 3. Nature of DNA Lesions and the Mechanism of DNA Repair

### 3.1. DNA Damage and DNA Repair are the Sine Qua Non of Radiation Injury

RT causes DNA damage in target tissues either directly by inducing DNA strand breaks or indirectly by generating reactive oxygen species (ROS) that oxidize proteins and lipids and thereby induce DNA strand breaks. RT yields different types of DNA damage, including abasic sites, single-strand breaks (SSBs) and double-strand breaks (DSBs). The spatial patterns of these areas of DNA damage vary based on the ionization track of the incident radiation: the damage is more dispersed and uniform with X- or γ-ray (low-LET) radiation and more discrete, clustered, and heterogeneously scattered along the beam tracks with charged particles (high-LET). DNA damage repair is an evolutionarily conserved process that maintains genomic integrity and repairs DNA lesions that occur naturally during routine cellular functioning or as a result of environmental insults. The cell’s repair capacity is mediated by a number of pathways that work in unison to repair individual DNA damages. Repair is intricately coupled with the blockade of cell cycle progression (at checkpoints), so that genomic integrity is restored before DNA replication and cell division.

DSBs are the most lethal DNA lesions if they go unrepaired. The key processes for repairing DSB damage are (i) non-homologous end joining (NHEJ), an error-prone process that joins broken ends with little or no additional processing before ligation, and (ii) homologous recombination (HR), in which loose ends are resected and homologous sequences from an undamaged sister chromatid are used as templates for new DNA synthesis. NHEJ is the predominant pathway for repairing simple DSBs induced by X- or γ-ray/photon beams [64], because it is active throughout the cell cycle. In contrast, HR is active only after DNA replication in the S and G2 phases, when a sister chromatid is present [65,66,67]. Thus, DSB repair pathway choice is decided not only by the cell cycle stages but also by the ionization density of the RT.

### 3.2. Charged Particles Induce Clustered DNA Double-Strand Breaks

High-LET charged particles, including ^12^C particles, induce clustered DNA damage, a unique class of DNA lesions that includes two or more individual lesions within one or two helical turns of the DNA, caused by the passage of a single radiation track [68,69,70]. There are two basic types of DNA lesions within clustered DNA damage: double-strand breaks (DSBs) and non-DSBs, which include abasic sites (apurinic/apyrimidinic sites or APs), damaged bases (oxidized purines or pyrimidines), and single-strand breaks (SSBs) [71]. Clustered DNA damage can be a mixture of DSBs, modified bases and SSBs. Evidence suggests that a major fraction of DSBs induced by high-LET radiation are considered clustered DSBs or are associated with other DSBs [72,73]. Furthermore, experimental evidence supports the notion that clustered DSBs induced by charged particles are more difficult to repair than isolated DNA lesions induced by X-rays [74,75,76]. For this reason, the relative biological effectiveness (RBE) for cell killing increases as the LET of the ionizing radiation increases.

### 3.3. Non-Homologous End Joining (NHEJ) is Required for Processing Simple DSBs

Error-prone NHEJ is a DSB repair mechanism that involves the Ku heterodimer, which consists of Ku70 and Ku80. This dimeric protein binds to the two ends of the DSB to recruit DNA-PKcs and Ligase4-XRCC4 to join the termini of the broken strands. NHEJ is also a conserved mechanism that is essential for variable, diversity and joining recombination for B and T lymphocyte development. Repair of low-LET radiation-induced DSBs follows fast kinetics and is largely mediated by the canonical NHEJ pathway, whereas high-LET IR-induced DNA damage repair is a slower process. Evidence indicates that NHEJ-specific enzyme-deficient cells are less sensitive to high-LET radiation-induced DNA damage than to low-LET radiation [76]. Another study revealed that NHEJ works with a different subset of proteins in complex DSB repair than in simple DSB repair [77]. In this context, the combined role of human protein Artemis, which is essential for immunoglobulin isotype determination (VDJ recombination), and ATM is important, as they promote DSB repair in the G1 and G2 phases by NHEJ and HR, respectively [72,78]. In addition, Artemis’ endonuclease function facilitates end resection and DNA repair. Since small DNA fragments deactivate Ku dimer and DNA-PKcs, Moore et al. have proposed a PARP-dependent alternative mechanism in NHEJ to explain the increased dependence on HR for repairing high-LET radiation-induced DSBs [78]. Thus, the NHEJ pathway plays a predominant role in repairing DSBs induced by low-LET radiation.

### 3.4. The Homologous Recombination Repair Pathway is Required for Processing Clustered DSBs

HR uses the sister chromatid or homologous DNA strand as a template to recover genetic information required to precisely repair DSBs in the S and G2 phases of the cell cycle. Damages that occur during the late S or G2 phase may be repaired by both HR and NHEJ pathways. The size and complexity of DNA lesions inflicted by high-LET IR determine whether the cells will be repaired by NHEJ or HR. Mostly, the slow DSB repair process that follows exposure to high-LET radiation is mediated by HR [79]. The recruitment of the MRN complex at the DSB ends initiates HR, then the protein complex MRE11/CtIP resects the 3′ end of the DSBs, which generates free ends to be extended by the 3′-5′ exonuclease activity of EXO1. The recruitment of the single-strand DNA binding protein 1 (hSSB1) and replication protein A (RPA) prevents further degradation of the 3′ free ends and also protects DNA from inappropriate genomic rearrangements and hairpin-like structure formation [80,81]. This RPA-mediated DNA end protection event is followed by the recruitment of the RAD51-BRCA2 complex, with five additional protein complexes (RAD51B-RAD51C-RAD51D-XRCC2-XRCC3), during which all RAD51/paralogs bind eight BRC domains of BRCA2 by replacing RPA [82]. The RAD51/BRCA2 complex is involved in finding the homologous sequence in the sister chromatid and aligning that sequence to facilitate strand exchange to bridge DSBs by DNA polymerase delta, resolvases (MUS81), endonuclease EME1, and the Holliday junction 5′ endonuclease GEN1. After DNA synthesis at the repair site is complete, the DNA ends are joined by DNA ligase1 (Figure 1). Recent evidence suggests that the HR pathway is required to repair clustered DSBs [76,83,84,85,86] and that cells defective in the HR pathway are sensitive to ^12^C beams and have elevated levels of persistent DSBs [76]. HR’s mechanism is quite well known, but the role of other HR factors in repairing clustered DNA damage needs further investigation.

DNA damage repair pathway factors are known to modulate immune signaling either by sensing DNA in the cytoplasm, promoting micronuclei accumulation or by releasing fragmented self-DNA to the cytoplasm [87,88]. For example, it has been shown that nuclear-derived self-DNA accumulate in the cytoplasm in the absence of RAD51 and that triggers STING-mediated innate immune signaling in response to high-LET radiation [86]. Additionally, a recent study found that NBS1, a DNA damage sensing factor, together with its binding partners regulate cGAS binding to micronuclei in response to genotoxic stress, including ionizing radiation [89]. Thus, failure to properly repair genomic DNA can generate both fragmented DNA and micronuclei, which in turn can trigger cytosolic DNA sensing pathway-mediated immune signaling (Figure 1).

### 3.5. Repairing Non-DSBs in Clustered DNA Damage: Base Excision Repair (BER)

BER is a major pathway for repairing base damages belonging to non-DSB lesions in clustered DNA damage sites characterized by the presence of two or more bistranded or tandem lesions within one or two turns of the DNA helix. SSBs, abasic sites, and oxidized purines and pyrimidines fall under the category of non-DSB lesions induced by ionizing radiation [90]. Using biophysical modeling, researchers have revealed a positive correlation between the complexity and occurrence of non-DSB clustered damage and the ionization density of the incident radiation. Moreover, the repair efficiency of non-DSB clustered damages is lower than that of isolated non-DSBs. The major factors that determine the repair efficiency are the type of cluster, the distance between the lesions, the number of lesions, and the relative orientation of the lesions. Reports of impaired SSB repair by a nearby SSB or AP site also support the above finding. Unlike isolated SSBs that are repaired by short-patch BER, clustered non-DSB sites are mainly repaired by long-patch BER. Moreover, the generation of persisting repair intermediates, such as DSBs, makes the process of repairing non-DSBs by BER very complicated, as shown by many standard molecular biological and biochemical studies [91,92].

## 4. Radiotherapy and Immune Signaling

The immune system acts as a vital player that recognizes and eradicates malignant cells via immunoediting, a dynamic process by which the activation of innate and adaptive immune responses determines a tumor’s fate. When all the constituents of the immune system work in synchrony, transformed malignant cells are effectively and efficiently eliminated. Occasionally, cells that evade elimination in this manner morph into an equilibrium state between elimination and overt escape. Tumors that break past this stage can escape immune surveillance and thrive and grow in an immunosuppressive tumor microenvironment.

RT has the potential to counteract this progression and trigger antitumor immunity. RT’s immunomodulatory effects appear to start with the induction of genotoxic stress—via defects in DNA repair, replication fork processing, telomere maintenance, and cell cycle progression (G2-M phase), among other causes—that can result in the accumulation of fragmented self-DNA and micronuclei. These self-DNA fragments and micronuclei appear as cytosolic DNA both in tumor cells and in immune cells that internalize tumor cell DNA fragments from the tumor microenvironment (TME) [93,94]. Although the immunomodulatory effects of X-ray RT have been extensively studied in vitro and in vivo in animal models and in patients, there is a paucity of preclinical and especially clinical data showing the impact of ^12^C particles on immune signaling [95,96,97,98].

Recent evidence suggests that DNA damage repair pathway factors, previously thought to function only in DNA damage sensing and repair, may also control signaling pathways in the innate immune system (Figure 2) [87]. These factors work directly or indirectly to repress cytosolic DNA sensing pathway-mediated immune signaling by masking cytosolic DNA. This negative regulation of the immune system helps to maintain the proper immune microenvironment in normal cells to prevent unnecessary or defective activation. Therefore, the immune system can modulate either tumor suppression or progression, and RT has the potential to regulate immune responses to yield antitumorigenic effects.

RT is a keystone for cancer treatment. However, its use is limited by radiation-induced toxicities. If these toxicities could be reduced, higher doses of radiation could be given to patients, which would facilitate a better cure. Ultra-high Dose rate (FLASH)-RT involves the ultra-fast delivery of a large single dose of radiation (10–20 Gy) at a mean dose rate above 100 Gy per second, which is several orders of magnitude greater than what is currently used in routine clinical practice [99]. Even more so than the mean dose rate, the instantaneous dose rate within a radiation pulse may be a key driver of FLASH effects. Several recent studies have demonstrated that FLASH therapy induces fewer toxicities in normal tissue than conventional RT [100,101,102]. The mechanism of FLASH therapy is still not very clear, but several hypotheses have been proposed. Some groups have suggested that differential responses between FLASH-RT and conventional RT may be caused by the depletion of oxygen at high doses of radiation [103,104]. In fact, multiple studies have found that FLASH therapy can deplete local oxygen and induce a short-lived protective hypoxic environment within normal healthy tissues that increases radioresistance. FLASH therapy may also modulate immune responses, which could contribute to its effect, but this needs more research [105]. FLASH-RT may directly affect immune cells or indirectly influence the tumor microenvironment. Some evidence suggests that immune cells are not preferentially spared by ultra-high dose rates similar to FLASH [106]. TGF-β is a master regulator of immune homeostasis following RT, largely mediating immunosuppressive effects. One experimental finding implies that TGF-β might be one of the key players regulating FLASH effects [102,107]. Macrophages also play important roles in radiation-induced pulmonary fibrosis, which FLASH-RT helps to reduce [102]. FLASH-RT’s effects on immunogenic cell death remain unknown. FLASH is an exciting treatment strategy that could change the future of clinical cancer treatment. Clinical trials and future research on FLASH therapy are necessary to promote its use for curing cancer in a single fraction with less toxicity [108].

To improve the efficacy of radiotherapy, it is important to understand the mechanisms of radioresistance. RT mainly affects and damages DNA, but upregulation of the DNA damage response is associated with radioresistance [109,110]. Mitotic cells are hypersensitive to radiation because they inactivate DSB repair [111], though B16-BL6 and PANC-1 cells have shown high radiosensitivity during the early S-phase [112]. We know that some tumors are more radiosensitive than others, but little is known about the roles that the immune response plays across a range of radiosensitive and radioresistant cells. One study reported that radiation reprogrammed the tumor microenvironment in the parental tumor, but resistant tumors were not affected much. The parental tumor showed more CD8^+^ T cell infiltration than resistant cells. Along with that, T cell chemokines that play an important role in the immune response were also more present in parental cells than in radioresistant cells. This raises the possibility that CD8^+^ T cell infiltration after radiotherapy might be related to tumor radiosensitivity in vivo [113]. Thus, the radioresistance and radiosensitivity of tumors are important parameters that determine the extent to which radiation activates the immune response. Radiation initiates immune responses by fragmented DNA and tumor cell death. Cancer cells that are resistant to radiation will have less of an immune response than radiosensitive cells.

### 4.1. Radiation and Innate Immune Signaling

RT can either damage DNA directly or create free radicals that affect DNA indirectly. Innate immune responses to DNA damage induced by radiation are mediated through cytosolic DNA sensors, innate immune receptors that sense double-stranded DNA. Cyclic GMP-AMP synthase (cGAS) is one of the most important DNA sensors that detects cytosolic DNA and triggers the expression of inflammatory genes that activate defense mechanisms. Tumor-derived DNA, such as micronuclei, DNA of dead tumor cells, cytoplasmic fragments, and free telomeric DNA, can activate the cGAS pathway and induce cellular senescence and antitumor immunity [88,114,115,116,117,118,119,120,121]. The progression of the cell cycle through mitosis after RT-induced DNA DSBs leads to the formation of micronuclei, cytoplasmic aggregates of damaged DNA encircled by a defective nuclear envelope. Breakdown of the micronuclear envelope exposes DNA to the cytosol, which activates the cytosolic DNA sensor cGAS and the downstream signaling effector stimulator of interferon genes (STING) pathway, which leads to the production of type I interferons (IFNs) [93,122,123,124,125,126] (Figure 3).

The cGAS pathway can also be deliberately activated as a therapeutic strategy to enhance the efficacy of radiation for treating tumors. In human fibroblast cells, extrachromosomal telomere repeats translocated to the cytoplasm and were detected by cGAS to initiate the STING signaling pathway to mediate the phosphorylation of interferon regulatory factor 3 (IRF 3), which results in IFN expression [121]. STING also plays important roles in autoimmune diseases initiated by aberrant cytoplasmic DNA [93]. Administering cGAMP, a second messenger and activator of STING, enhances the antitumor immunity induced by radiation.

Several studies have also indicated that, in the TME, expression of type I interferon beta (IFN-β) in dendritic cells (DCs) depends on STING and is modulated by tumor-derived antigens [114]. Thus, during cGAS pathway activation after radiation, IFN-β is produced, which recruits and activates DCs that present antigens to CD8 T cells (Figure 3). This stimulus is a prerequisite for the priming of CD8^+^ T cells and antitumor immunity. Activating AIM2 (absent in melanoma 2), a cytoplasmic DNA sensor that recognizes double-stranded DNA induced by radiation, causes the inflammasome to assemble, which activates caspase1; this, in turn, leads to the release of proinflammatory cytokines and promotes cell death [127,128]. Thus, AIM2 can be used as a therapeutic target for cancer patients receiving RT.

Recent work by Vanpouille-Box et al. has identified DNA exonuclease Trex1 as an upstream regulator of radiation-induced antitumor immunity. They have demonstrated that Trex1 expression depends on the radiation dose. When radiation is given in a single, high dose (>12 Gy), Trex1 is expressed at high levels and degrades the accumulated DNA in the cytosol of cancer cells, thus preventing type I IFN activation mediated by the cGAS pathway. In contrast, when radiation is given in multiple, low-dose fractions, Trex1 induction is prohibited, and it stimulates cancer cells to produce IFN and mediate tumor regression. If Trex1 is not induced, it leads to the priming of tumor-specific CD8^+^ T cells that, in the presence of immune checkpoint inhibitors, mediate complete durable regression of irradiated and non-irradiated tumors [94].

The combination of a STING activator with image-guided RT synergistically controlled both local and distant pancreatic cancer in mouse models [129]. In contrast, several reports suggest that, in some models, STING can also drive immunosuppression. One study reported that STING activity induced by indoleamine 2,3-dioxygenase promoted the growth of Lewis lung carcinoma [130]. In another report, radiation-induced STING and type I IFN recruited myeloid-derived suppressor cells (MDSCs) to irradiated tumor cells by the chemokine receptor 2 (CCR2) pathway and caused immunosuppression and radioresistance. This effect of STING agonists can be abrogated by administering a CCR2 antibody [131]. All of these studies indicate that innate immune responses can be activated by RT.

### 4.2. Radiation and Adaptive Immune Signaling

Radiation can also modulate immune signaling in the tumor and alter the TME. In doing so, RT can induce antitumor adaptive immunity, which leads to tumor control [93,132,133,134,135]. Innate and adaptive immunity work together to generate a systemic immune response. In cancer, radiation induces DNA damage that activates innate immune responses, and tumor cell death exposes tumor antigens that act as further targets for adaptive immune responses.

Antigen exposure and presentation is the first step in initiating the immune signaling pathway. Antigen presenting cells (APCs) engulf the tumor cells and present their antigens to T cells via phagocytosis to activate immune signaling (Figure 3). RT promotes the translocation of calreticulin, a calcium-binding protein that promotes phagocytosis, from the endoplasmic reticulum to the plasma membrane [136]. In addition, radiation downregulates proteins that trigger the antiphagocytosis signal CD47 [137]. Thus, radiation enhances APCs’ clearance of damaged tumor cells and enhances T cell priming.

Recent studies have also revealed that radiation-induced viral mimicry can stimulate robust tumor-specific CD8 T cell responses that can mediate systemic tumor regression in conjunction with immunotherapy [94,138]. Radiation mediates tumor regression by the type I IFN pathway and the adaptive immune response. One study reported that STING induces type I IFN by intracellular exogenous DNA independently of toll-like receptors (TLRs) [139]. Type I IFN activates DCs and contributes to T cell priming. Moreover, type I IFN directly kills tumor cells and inhibits their proliferation. This has been shown by reports on administering exogenous IFN to treat leukemia [140].

TLRs are major players in innate immunity that are usually expressed in APCs and that recognize the conserved molecules of different microbes, the so-called pathogen-associated molecular patterns (PAMPs). Secreted HMGB-1 (high mobility group box 1), a chromatin-associated nuclear protein, binds to TLR-4 in response to chemotherapy and targeted radiotherapy and contributes to antitumor effects [132].

Radiation also elicits signals that stimulate TLR-4 on DCs, which suggests another way by which RT modulates immune responses. Dying cells secrete danger signals, so-called damage-associated molecular patterns (DAMPs), that aid in DCs’ maturation and activation. Radiation can induce surface exposure or release of these danger signals and activate immune cells. After the phagocytosis of tumor cells, DCs present tumor antigens to T cells through MHC molecules, which results in the priming and activation of T cells. T cells are activated by two pathways: the first signal is initiated by the binding of the MHC-antigen complex and antigen-specific T cell receptors (TCRs), and the second signal is delivered via costimulatory molecules expressed on the cell surface of activated APCs and cytokines produced either by the APC or by the activated CD4 T cell itself [141]. DCs that are exposed to DAMPs upregulate the expression of costimulatory molecules and initiate the tumor specific T-cell response [136] (Figure 3). Thus, RT enhances the immune system’s activity against tumors by multiple mechanisms. These T-cell mediated local effects of radiation could stimulate a robust systemic immune response that affects local antitumor responses within tumors distant from the irradiated site.

### 4.3. Radiation and Cancer Vaccines

Cancer vaccines are a form of immunotherapy that induces an immune response specific to a particular type of cancer to treat it or prevent its development. For example, Sipuleucel-T, a dendritic cell-based vaccine, has shown effectiveness in treating advanced prostate cancer and has been approved by the FDA [142]. The emerging combination of cancer vaccines with RT is a promising treatment option. RT induces immunogenic tumor cell stress and cell death, which enhanced sensitivity to T cells and synergized with cancer vaccines in preclinical studies [143,144]. Several studies have described cancer vaccines’ potential to expand the adaptive immune response to radiation [143,145,146]. One of these studies found that radiation releases tumor-associated antigens (TAAs) like carcinoembryonic antigen (CEA) and mucin-1 in carcinoma cell lines and generates tumor-specific T cells [142].

Radiation modifies the phenotype of tumors to render them more susceptible to vaccine-mediated T cell killing and alters the microenvironment to promote greater infiltration of immune cells [147,148,149,150,151]. In 2009, Lee et al. reported that ablative RT increases T cell priming in draining lymphoid tissue and reduces the growth of primary and metastatic tumors [133,152]. CD8 T cells use their T cell receptors to recognize tumor-derived antigens bound to MHC I and to attack tumor cells. Radiation plays an important role in increasing the peptide repertoire and MHC I expression on tumor cells and boosts the efficacy of immunity in vivo.

Several mechanisms of radiation-induced immunomodulation have been reported. One is the upregulation of Fas (CD95) on hematologic and non-hematologic cells [153]. Fas is a member of the tumor necrosis factor (TNF) receptor family that induces apoptosis upon ligation with a Fas-ligand. It also plays important roles in the immune system, including an apoptotic selection process during T cell development, clonal deletion of autoreactive T cells in the periphery and as an effector of cytotoxic T lymphocytes (CTLs) [154,155]. CD8 CTLs mediate the lysis of targets via one of two mechanisms after MHC/antigen recognition: by secreting perforin and granzyme contents or by triggering the docking of Fas with FasL expressed by the activated CTL [156,157]. Chakraborty et al. found that upregulated Fas on tumors was solely responsible for enhancing the efficacy of vaccine therapy in their model, as tumors could not regress in tumor cells defective in Fas signaling [147]. Thus, radiation can play a critical role in improving the effectiveness of anticancer vaccines by modulating immunity.

Administering cancer vaccines before or together with the radiation dose may be ineffective because the tumor site lacks an intact adaptive immune system. Thus, it is better to administer vaccines after RT, using the vaccine as a booster for the immune cells generated by RT rather than having RT deplete the infiltrating immune cells secondary to vaccine therapy. Furthermore, RT enhances the secretion of chemokine CXCL16 by human and mouse breast cancer cells, which attracts and recruits CD8-positive effector T cells expressing the CXCR6 receptors [158]. It is not clear how long this effect lasts, however, because that experiment was extended only to 72 h after irradiation. Nonetheless, these findings provide a rationale for administering vaccines after RT to increase its efficacy.

TLR agonists are important candidates for use as cancer vaccines because they bridge innate and adaptive immunity through a different pathway: cytokine release and IFN production by DCs and macrophages. Radiation in combination with these TLR agonists enhances the availability and the presentation of TAA. This was confirmed by a study that administered TLR 7 agonist R848 and RT to mice affected by B cell and T cell lymphoma and found that the combination led to better tumor clearance than either treatment alone [159]. A clinical trial tested the combination of TLR agonist and RT on 15 patients with mycosis fungoides, a subtype of cutaneous T cell lymphoma, and monitored distant untreated sites for systemic responses. Five patients responded to treatment and showed reduced levels of Tregs [160]. Thus, clinical and preclinical data show that TLR agonists can be a powerful tool in priming radiation as an in situ cancer vaccine. Other data with the classical TLR9 agonist, CpG oligodinucleotides (ODN), suggest that combining RT with intratumoral administration of CpG ODN and anti-CTLA-4 and anti-OX40 antibodies overcomes the tolerogenic influence of Treg cells in many aggressive cancers, stimulates a tumor-specific systemic immune response, and eradicates distant metastases, whereas systemically administered antibodies are less effective in such combinations [161].

Viral vaccines use viruses to potentiate the patient’s immune system against tumors. One preclinical study used a combination of RT and recombinant fowl pox vectors to treat a colon cancer animal model transfected with CEA. The tumors treated with this combination demonstrated a massive infiltration of T cells not seen with either modality alone [147]. This study demonstrates a new hypothesis for using RT and vaccines in combination to stimulate antitumor responses against existing tumors [137,162,163,164,165,166,167].

Radiation’s unique ability to increase tumor antigen presentation makes it a promising modality to combine with chimeric antigen receptor (CAR) T-cell therapy. In CAR T-cell therapy, T cells are isolated from patients and genetically engineered to express CAR, which recognizes tumor antigens. Later, CAR T cells are infused into the patient to induce tumor cell death. The use of CAR T cells in solid tumors presents the unique challenge of effectively trafficking the genetically engineered T cells to the target and propagating its response in an adverse tumor microenvironment. The rich stroma of solid tumors creates a physical barrier against the penetration and aggregation of engineered T cells, which lack the enzyme heparinase [168]. Even after the T cells have penetrated, molecular signaling in the tumor microenvironment is immunosuppressive, and it downregulates the recruitment of additional T cells [169,170,171,172]. Radiation could enhance MHC-I expression and antigen presentation to make tumors more feasible targets of CAR T cells [173].

Despite the vast body of research on the systemic immune response after RT, especially in combination with immunomodulatory drugs, there is a lack of data directly examining the synergy between radiation and CAR T-cell therapy. However, several preclinical studies have revealed the mechanism by which RT initiates tumor-specific immune responses, which can provide the basis for overcoming some of the challenges facing CAR T-cell therapy in solid tumors. High expression of tumor antigen as compared to the normal tissue is very important for this therapy. RT has the potential to increase tumor-associated antigens and cell surface receptors. One study reported that RT led to a dose-dependent increase in the expression of MHC1 molecules on the surface of tumor cells [174]. We have mentioned above that RT releases tumor-associated antigens like CEA, so it is interesting that CAR T cells directed against CEA are in a clinical trial. Mesothelin is another target of interest in a study of CAR T-cell therapy for lung cancer. One study observed that mesothelin expression per cell was higher in the irradiated group than in the control group [175]. RT can target other barriers to CAR T-cell therapy, such as the trafficking of CAR T cells into solid tumors. RT can upregulate adhesion molecules like ICAM1 and VCAM1 to promote adhesion [176,177]. Thus, even as the role of RT in CAR T-cell therapy is still evolving, we can say that RT would be a strong adjunct for this therapy.

### 4.4. Radiation and Tumor Antigens

Tumor antigens present another opportunity to leverage radiation’s immunogenic properties to enhance the antitumor effects of combination therapies. Tumor antigens may be divided into two categories based on their pattern of expression: tumor-associated antigens (TAAs) and tumor-specific antigens (TSAs). TAA are the proteins that are shared between that particular tumor and its normal tissue but distinct from other tissues. On the contrary, TSA arise from tumor-specific mutation, which result in exclusive expression of that particular antigen in the tumor that makes it more specific. TSAs are widely known as neoantigens, which is the repertoire of peptides that display on the tumor cell surface and could be specifically recognized by neoantigen-specific TCRs [178,179,180,181,182].

From an immunological point of view, a tumor neoantigen is a foreign protein that is completely absent from normal organs and tissues. Since neoantigens may be recognized by the host immune system as non-self, they are ideal targets for immunotherapy. TAAs and cancer-germline antigens (CGAs) are also found in healthy and immune-privileged tissues, which make them less useful for immunotherapy than neoantigens [180,183,184]. Several studies have suggested that neoantigens are highly immunogenic and play a crucial role in tumor-specific T cell-mediated antitumor immunity. Several phase I clinical trials have demonstrated the feasibility of treating melanoma and glioblastoma with neoantigen vaccines that induce neoepitope-specific T cells that kill autologous tumor cells [185,186,187]. Thus, interest in neoantigen-based vaccines is increasing rapidly, and several preclinical and clinical studies have shown their antitumor response.

Recent preclinical and clinical studies have shown that all neoantigen-based approaches can induce antitumor immune responses in individual tumor microenvironments. Overall, preclinical and clinical records have suggested that RT recruits DCs and CD8 cells and enhances tumor antigenicity by inducing a burst of gene transcription that is likely to generate many new and potential immunogenic peptides for loading onto MHC-I of both DCs and cancer cells. CD4 T cell responses that are specific to neoantigens exert their helper function at the level of DCs and enhance the activation of antitumor CD8 T cells [183]. The abundance of an antigen is critical to achieving an effective presentation via the endosomal APC pathway [188], so it can be assumed that the radiation-induced mutanome may boost neoantigen presentation by MHC-II, thereby enhancing the activation of CD4 T helper responses.

Vaccines based on TAAs or CGAs have not achieved convincing results because of the central and peripheral tolerance mechanisms [185]. In addition, TAAs or CGAs are present in some healthy tissues, so targeting them may result in severe autoimmune toxicities, such as hepatitis, colitis, renal impairment, and treatment-related death [189]. These data have important implications for choosing the correct radiation dose and interval of RT.

### 4.5. Trial Studies Combining RT with Immunomodulators

RT’s ability to induce tumor cell death and to counteract an immunosuppressive tumor microenvironment to effectively convert the irradiated tumor into an in situ vaccine has great implications for curing cancer, and it provides the primary basis for combining novel immunotherapies with RT. The combination of immunotherapy with RT is an actively growing field of clinical investigation, with a rapid expansion in the number and type of clinical trials.

Immunotherapy treatment includes immune checkpoint blockade, adoptive T cell transfer, cytokine therapy, dendritic cell and peptide vaccines, and monoclonal antibody treatment. Many of these immunotherapies have been tested in combination with RT in preclinical studies and are under investigation in the clinic.

CTLA-4 is an inhibitory receptor that negatively regulates T cell activation [190]. A phase 1 dose escalation trial (NCT01497808) evaluated 22 patients with metastatic melanoma who were treated with 2 and 3 fractions of RT to a single lung or osseous metastasis, or with 6 Gy to a subcutaneous or hepatic metastasis, followed by ipilimumab [191]. There were no complete responses, though partial responses were noted in 18% of unirradiated lesions. A phase 1/2 trial enrolled men with metastatic castrate-resistant prostate cancer and escalated the dose of ipilimumab, with or without concurrent single-fraction RT (8 Gy) targeting an osseous metastasis [192]. This approach was further evaluated in a subsequent phase 3 trial (NCT00861614), which found that patients with non-visceral metastatic disease treated with ipilimumab combined with RT experienced an incremental improvement over RT-alone in overall survival [193].

PD-1 is an inhibitory cell surface receptor that acts as an immune checkpoint. Its ligand, PD-L1, is expressed on diverse types of cells, including antigen presenting, epithelial, and endothelial cells. Studies have shown that PD-1 and PD-L1 targeted therapies have clinical activity against metastatic bladder cancer, head and neck cancers, Hodgkin’s lymphoma, non-small cell lung cancer, and renal cell cancer [194]. Preclinical studies have demonstrated that the combination of RT and targeted PD-1/PD-L1 therapy activates cytotoxic T cells, reduces myeloid-derived suppressor cells, and induces an abscopal response [195,196]. Based on these promising results, numerous ongoing clinical trials are testing the combination of PD-1/PD-L1 inhibition and RT. Many of these studies are phase 1 or 2. Two open phase 3 trials are looking at the combination of nivolumab with RT in locally advanced NSCLC (NCT02768558) and glioblastoma (NCT02617589).

GF-β is a cytokine with immunosuppressive activity that is activated by RT in the tumor microenvironment [197]. Preclinical studies have shown that inhibiting TGF-ß during and after RT allows the priming of T cells to multiple tumor antigens, which leads to immune-mediated regression of the irradiated tumor and metastatic tumors [198]. A few ongoing clinical studies are testing the benefits of TGF-β inhibition and RT. One phase 1/2 trial in patients with metastatic breast cancer tested the use of a neutralizing antibody, fresolimumab, with RT (NCT01401062). Another phase 2 study is testing LY2157299 in combination with chemotherapy and RT for rectal cancer (NCT02688712).

Dendritic cell (DC)-based immunotherapy involves the intratumoral injection of autologous DCs to promote the cross priming of T cells to tumor antigens after RT and to increase the tumor infiltration of CD8^+^ T-cells [199]. This approach has been utilized in a phase 1 trial in patients with hepatocellular carcinoma who were treated with single-fraction RT (8 Gy) and injected with autologous DCs two days later. Fifty percent of 14 patients showed a minor to partial clinical response [200]. Another phase I study (NCT00365872) evaluated the immunologic response to intratumoral DC injection with neoadjuvant RT to 50 Gy in 25 fractions in soft tissue sarcoma [201]. Ten of 18 patients (56%) exhibited a tumor-specific immune response. The number of clinical trials exploring the use of RT with immunotherapy is rapidly increasing. More research is needed to optimize the conditions to make this combination more effective in the clinic.

## 5. Radiotherapy and Abscopal Effects

RT primarily treats localized tumors, but it can also inhibit distant metastatic tumors after local radiation therapy, which is called the abscopal effect (Figure 2). Although the exact mechanism that yields the abscopal effect is not yet fully understood, several studies have partially elucidated how combining radiotherapy with immunotherapy could improve this effect.

The major effects of radiation are its cytotoxic effects, which result from direct DNA damage and indirect generation of cell-damaging free radicals. These effects are confined to the tumor receiving the RT. RT not only damages the tumor, but it also modulates the TME to exert an antitumor immune response. This immunological response releases different cytokines and chemokines into the tumor environment, which causes chemoattraction and recruitment of DCs to the site of the tumor. Activation of DCs and upregulation of cytotoxic T cells have been reported as mechanisms of the abscopal effect. The combination of RT and immunotherapy induces upregulation of tumor-related antigens in MHC-I so that it can activate more CTLs. This effect can be exploited by combining RT with immunotherapy to treat radiation-resistant tumors and to boost the antitumor responses of adoptively introduced tumor-specific CTLs. These CTLs then recognize the radiation-induced antigenic peptides or enhanced tumor-related self-antigens to elicit a better tumor response. Radiation is applied locally, so taking advantage of the specificity of both RT and immunotherapy can add a new perspective to existing cancer treatments. Thus, radiation therapy boosts the antitumor immunity of a wide range of immunotherapies, including checkpoint inhibitors and adoptive transfer of T cells [174,202,203].

Many researchers have also validated that RT can also cause local bystander effects, where the treated tumor cells influence neighboring cell properties or vice versa through varied mechanisms [204,205]. In such cases, irradiated cancer cells may release signals that can influence nearby non-irradiated cells. In contrast, the abscopal effect is a long distance and systemic effect at a distant metastatic site, which is usually mediated by immune cells such as T cells. Furthermore, RT releases cytokines, which have been shown to play a vital role in the abscopal effect [206]. Bystander effects are mediated by several mechanisms, like direct gap-junction mediated cell–cell communication via ions such as calcium or nitric oxide, though immune factors like transforming growth factor-β and cytokines can also be released in the extracellular compartment and trigger local immune activation [207]. Macrophages also play an important role in the bystander effect: once they are activated by radiation, they damage neighboring cells by transferring different signaling factors [208,209,210]. Thus, both abscopal and bystander effects are generated by radiation-mediated immunological responses. However, bystander effects are triggered by other pathways and signaling mediators.

Evidence indicates that the abscopal effect is immune-mediated and can be induced if radiation is combined with strategies that target tumor-associated immunological dysfunction [211,212,213]. Several preclinical studies have tested combinations of radiation with immune modulators for their ability to induce local and systemic antitumor T cell responses [214,215]. Some of these combinations have been translated into clinical trials too. Several researchers have elucidated the immunological effects of radiation, but there are still no methods available to predict how radiation will enhance immunological responses. We need to understand why some patients respond to radiotherapy alone or with immunomodulatory drugs and exhibit abscopal effects while others fail to do so. Some studies have tried to explain these failures by associating them with the host, the tumor or the treatment. Host criteria include hematological impairment at the time of combined treatment, neutrophil to lymphocyte ratio >4 [216], and the presence of a microbiome [217]. Tumor-specific causes include the downregulation of the molecular machinery required for IFN-I activation, induction of immunosuppressive mediators like PDL-1 and TGF, and antigenic heterogeneity in different metastases [195,218,219,220]. The dose and fractionation of RT can also influence the immune system, but the immunological effects of different regimens are unpredictable. Thus, it is important to identify effective biomarkers that can predict abscopal effects or show an immunological response.

An irradiated tumor cell can undergo one of several types of cell death, so the cellular stress or DNA damage and modifications in the tumors may lead to the liberation of neoantigens. These can stimulate a tumor-specific immune response when the neoantigens are engulfed by APCs and then presented to CD8-positive T cells. The CD8-positive T cells can then recognize and attack both the primary tumor and metastatic disease [221].

Overall, combining immunotherapy with RT holds great promise for eliciting abscopal responses. The problem lies in enhancing abscopal responses clinically in a more effective manner. We need to understand more about the parameters that predict abscopal effects and how to enhance RT-induced immune responses so we can use this phenomenon in the clinic to treat metastatic cancer.

## 6. Conclusions and Future Perspectives

RT has been widely used for years to treat various types of tumors. Technological advancements have fueled the development of sophisticated machines capable of delivering ionizing radiation precisely to tumors, but the fundamental mechanisms by which RT kills tumor cells are independent of the technique used for radiation delivery. Although RT’s effects on immune signaling have been known for years, our understanding of how RT modulates both innate and adaptive immune signaling is still evolving. In particular, a more nuanced understanding of the crosstalk between DNA damage caused by radiation, different DNA repair processes, and immune activation is crucial for advancing novel radiation-immunotherapy combinations that may elicit robust systemic immune responses. Modulating this crosstalk may convert the abscopal responses that are only sporadically observed with radiation–immunotherapy combinations into a reliable on-demand triggerable event. Towards this end, recent discoveries of new players in immune signaling and new paradigms in nuclear-to-cytoplasmic shuttling of chromatin and double-strand DNA fragments of radiation injury may provide clues for future interventions that are evidence-based and personalizable for individual tumors in specific clinical contexts. Identifying intrinsic and extrinsic factors that can modulate immune components within the TME and DNA repair factor(s) that can provoke the immune system in response to RT could benefit patients with treatment-resistant tumors. In turn, these targeted radiation interventions could define new paradigms where local treatment might be utilized to achieve systemic effects.

## Figures and Tables

**Figure 1 ijms-21-08151-f001:**
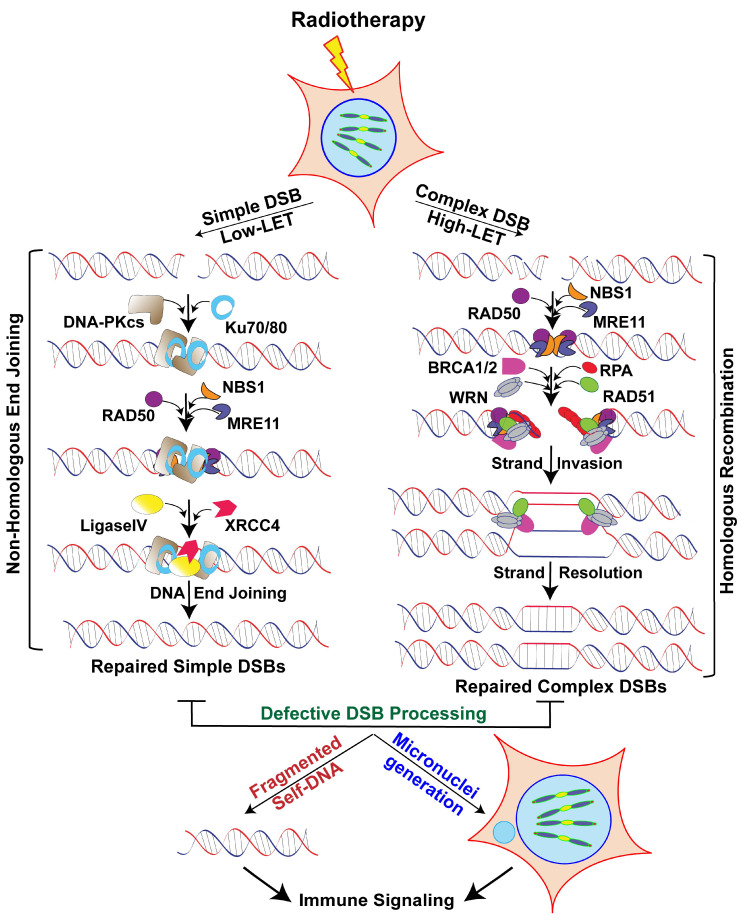
Schematics of the double-strand break (DSB) repair pathways after low- and high-linear energy transfer (LET) radiation. Low-LET radiation induces simple DSBs, which are repaired by non-homologous end joining pathway (NHEJ). The NHEJ process is initiated by the binding of a protein complex, the Ku70/80 heterodimer, to both ends of the broken DNA molecule, which, together with the catalytic subunit of the DNA-dependent protein kinase (DNA-PKcs), generates the DNA-PK complex. Non-ligatable DNA termini are processed by the MRN complex (Mre11/Rad50/Nbs1), and the DNA ends are later rejoined by ligase IV and XRCC4. High-LET radiation causes clustered DNA lesions (or multiply damaged sites). Clustered DNA damage is mostly repaired by homologous recombination (HR). HR is initiated by the resection of the DNA ends through the combined action of the MRN complex (MRE11, RAD50, and NBS1) and CtIP to generate single-stranded DNA overhangs. Subsequently, replication protein-A (RPA) binds to the newly created single-strand region before being exchanged with RAD51. The RAD51/ssDNA complex invades the homologous template DNA, creating a temporary triple-DNA structure in which strand exchange occurs, then DNA synthesis proceeds until the second end is captured. In subsequent steps, a Holliday junction is generated to prime DNA synthesis, which is further resolved by specific nucleases.

**Figure 2 ijms-21-08151-f002:**
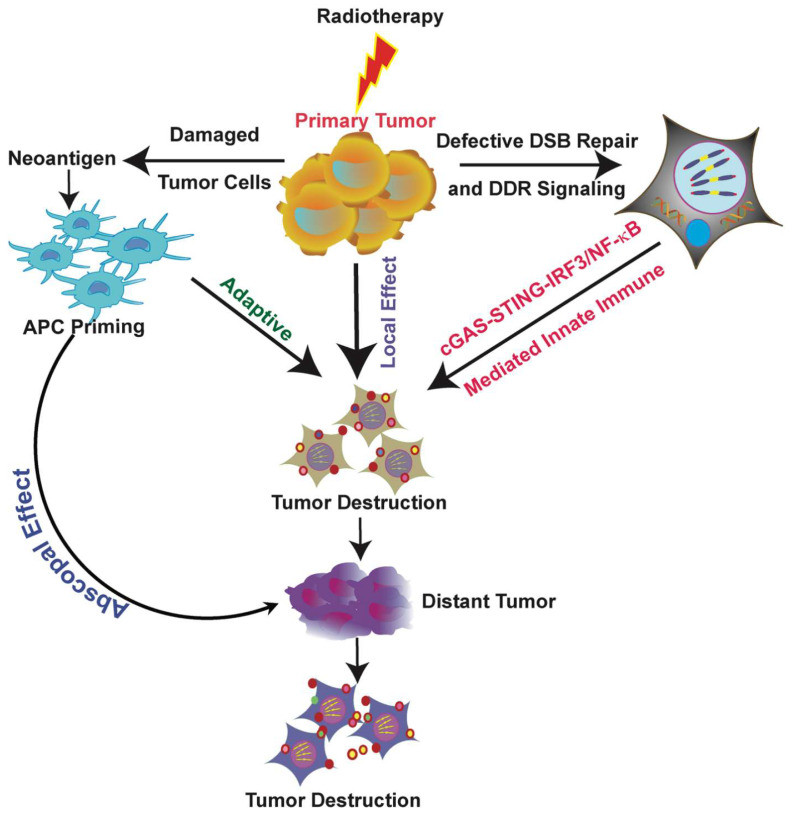
Schematic diagram of the comprehensive immunological effects of radiotherapy. DNA fragments generated by radiotherapy (RT) can trigger both adaptive and innate immune signaling pathways. RT causes double-strand DNA damage and destroys tumor cells. Tumor cell death exposes tumor antigens (neoantigens), which are specifically expressed on the tumor cell surface and activate antigen presenting cells (APC) like dendritic cells (DC). Primed DCs initiate adaptive immune response and kill tumor cells recognized by T cells, which play an important role in antitumor immunity. In addition, DNA damage releases fragmented DNA into the cytosol or generates micronuclei, which activates a cGAS-mediated innate immune response that destroys the tumor. RT primarily treats local tumors, but it can also inhibit distant tumors after local radiation, which is known as an abscopal effect. Thus, the systemic antitumor response to local treatment can target both local tumor cells and distant metastatic tumors.

**Figure 3 ijms-21-08151-f003:**
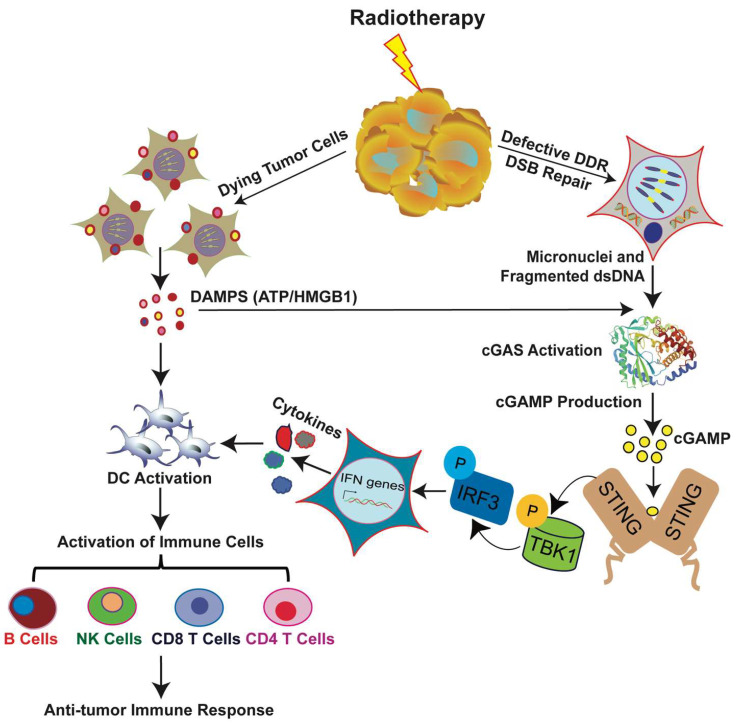
Schematic representation of a working hypothesis on the immunomodulatory role of radiation. Radiation induces DNA damage and generates micronuclei and fragmented dsDNA. During phagocytosis of tumor cells by dendritic cells (DCs), the DNA fragments hidden in irradiated tumor cells are released from phagosomes into the cytoplasm, where they act as a “danger signal”. The DNA sensor cGAS recognizes this signal, becomes catalytically active, and forms cGAMP. cGAMP binds to STING, which in turn activates IRF3 to induce type I IFN production. Type I IFN signaling activates DCs and promotes CD8^+^ T cell priming, which leads to tumor control. Alternatively, RT induces dying tumors to express more tumor antigens on their surface and to release danger-associated molecular patterns (DAMP), such as ATP and HMGB1, that stimulate DC activation. DCs express tumor antigen on their surface, interact with the T cell receptors on CD4^+^/CD8^+^ T cells, and activate immune cells that mediate tumor-specific killing.

**Table 1 ijms-21-08151-t001:** Advantages of Proton Therapy.

Cancer Type	Radiotherapy	Type of Study	Beneficial Effects	Reference(s)
Prostate	Proton	Trial study unknown	Better target delivery of radiation, low exit dose	[27,28,29]
				[30,31,32]
Head and Neck	Proton	Trial study unknown	Less radiation-induced tissue damage	[33,34,35]
			Better survival	[36]
			Decreased rates of tube feeding dependency, better quality of life	[37]
Non-Small Cell Lung Cancer	Proton vs. IMRT	Randomized study	Better dosimetric indices for heart sparing, and the risk of pneumonitis decreased over the duration of the study	[38]
Liver and Colon	Proton vs. SBRT	Trial study unknown	Longer survival than SBRT, spares more normal liver cells from radiation damage than treating with conventional RT	[39,40,41]
Esophageal	Proton vs. IMRT	Randomized Phase II	The mean total toxicity burden was considerably lower with protons than with IMRT	[42]
Brain (Adult)	Proton	Trial study unknown	Significantly reduced the side effects and better neurocognition over time after treatment	[43,44,45,46]
Brain (Pediatric)	Proton vs. Photon	Trial study unknown	Spared more surrounding normal tissue; reduced side effects and increased five-year survival rates (72–100%)	[32,47,48,49,50]
			Safer to deliver high dose of radiation and better neurocognition	[44,45,46,51]

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
