# Peer review of "Immunomodulatory Effects of Radiotherapy"

_ijms, 2020, doi:10.3390/ijms21218151_

Round 1
Reviewer 1 Report
The manuscript by Kumar and colleagues consists in a review paper on the Immunomodulatory Effects of ionising radiation. The paper is well written and well organised. In my opinion, this review could be of interest for a comparably large public, due to the rising interest on combined immune-radiation therapy.
Apart from the overall positive feedback on the manuscript, I would suggest the Authors expanding the section on Radiotherapy, eventually mentioning trial studies dedicated to the clinical investigation of radio-immune mediated effects. At the same time, it might be interesting to have a short comment on how immune signaling could be modulated by FLASH therapy, which is gaining increasing attention in the community.
At the same time, I would suggest shortening the initial description of radiotherapy modalities, which can be found elsewhere in more technical contributions.
Minor comment: please provide reference to LEM model at page 4.
Author Response
Our response to Reviewer 1 comments:
The manuscript by Kumari and colleagues consists in a review paper on the Immunomodulatory Effects of ionising radiation. The paper is well written and well organised. In my opinion, this review could be of interest for a comparably large public, due to the rising interest on combined immune-radiation therapy.
Comment #1: Apart from the overall positive feedback on the manuscript, I would suggest the Authors expanding the section on Radiotherapy, eventually mentioning trial studies dedicated to the clinical investigation of radio-immune mediated effects. At the same time, it might be interesting to have a short comment on how immune signaling could be modulated by FLASH therapy, which is gaining increasing attention in the community.
Our Response #1: We thank you for this suggestion. We have included the following paragraphs about the trial studies combining RT with immunomodulators in our revised version:
3.5 Trial studies combining RT with immunomodulators: RT’s ability to induce tumor cell death and to counteract an immunosuppressive tumor microenvironment to effectively convert the irradiated tumor into an in situ vaccine has great implications for curing cancer, and it provides the primary basis for combining novel immunotherapies with RT. The combination of immunotherapy with RT is an actively growing field of clinical investigation, with a rapid expansion in the number and type of clinical trials.
Immunotherapy treatment includes immune checkpoint blockade, adoptive T cell transfer, cytokine therapy, dendritic cell and peptide vaccines, and monoclonal antibody treatment. Many of these immunotherapies have been tested in combination with RT in preclinical studies and are under investigation in the clinic.
CTLA-4 is an inhibitory receptor that negatively regulates T cell activation [183]. A phase 1 dose escalation trial (NCT01497808) evaluated 22 patients with metastatic melanoma who were treated with 2-3 fractions of RT to a single lung or osseous metastasis, or with 6 Gy to a subcutaneous or hepatic metastasis, followed by ipilimumab [184]. There were no complete responses, though partial responses were noted in 18% of unirradiated lesions. A phase 1/2 trial enrolled men with metastatic castrate-resistant prostate cancer and escalated the dose of ipilimumab, with or without concurrent single-fraction RT (8 Gy) targeting an osseous metastasis [185]. This approach was further evaluated in a subsequent phase 3 trial (NCT00861614), which found that patients with non-visceral metastatic disease treated with ipilimumab combined with RT experienced an incremental improvement over RT-alone in overall survival [186].
PD-1 is an inhibitory cell surface receptor that acts as an immune checkpoint. Its ligand, PD-L1, is expressed on diverse types of cells, including antigen presenting, epithelial and endothelial cells. Studies have shown that PD-1 and PD-L1 targeted therapies have clinical activity against metastatic bladder cancer, head and neck cancers, Hodgkin's lymphoma, non-small cell lung cancer, and renal cell cancer [187]. Preclinical studies have demonstrated that the combination of RT and targeted PD-1/PD-L1 therapy activates cytotoxic T cells, reduces myeloid-derived suppressor cells and induces an abscopal response [188, 189]. Based on these promising results, numerous ongoing clinical trials are testing the combination of PD-1/PD-L1 inhibition and RT. Many of these studies are phase 1 or 2. Two open phase 3 trials are looking at the combination of nivolumab with RT in locally advanced NSCLC (NCT02768558) and glioblastoma (NCT02617589).
GF-ß is a cytokine with immunosuppressive activity that is activated by RT in the tumor microenvironment [190]. Preclinical studies have shown that inhibiting TGF-ß during and after RT allows the priming of T cells to multiple tumor antigens, which leads to immune-mediated regression of the irradiated tumor and metastatic tumors [191]. A few ongoing clinical studies are testing the benefits of TGF-ß inhibition and RT. One phase 1/2 trial in patients with metastatic breast cancer tested the use of a neutralizing antibody, fresolimumab, with RT (NCT01401062). Another phase 2 study is testing LY2157299 in combination with chemotherapy and RT for rectal cancer (NCT02688712).
Dendritic cell (DC)-based immunotherapy involves the intratumoral injection of autologous DCs to promote the cross priming of T cells to tumor antigens after RT and to increase the tumor infiltration of CD8+ T-cells [192]. This approach has been utilized in a phase 1 trial in patients with hepatocellular carcinoma who were treated with single-fraction RT (8 Gy) and injected with autologous DCs two days later. Fifty percent of 14 patients showed a minor to partial clinical response [193]. Another phase I study (NCT00365872) evaluated the immunologic response to intratumoral DC injection with neoadjuvant RT to 50 Gy in 25 fractions in soft tissue sarcoma [194]. Ten of 18 patients (56%) exhibited a tumor-specific immune response. The number of clinical trials exploring the use of RT with immunotherapy is rapidly increasing. More research is needed to optimize the conditions to make this combination more effective in the clinic.
Comment #2: Apart from the overall positive feedback on the manuscript, I would suggest the Authors expanding the section on Radiotherapy, eventually mentioning trial studies dedicated to the clinical investigation of radio-immune mediated effects. At the same time, it might be interesting to have a short comment on how immune signaling could be modulated by FLASH therapy, which is gaining increasing attention in the community.
Our Response #2: We thank you for this suggestion. We have included the following text about the FLASH therapy in our revised version:
RT is a keystone for cancer treatment. However, its use is limited by radiation-induced toxicities. If these toxicities could be reduced, higher doses of radiation could be given to patients, which would facilitate a better cure. FLASH-RT involves the ultra-fast delivery of a large single dose of radiation (10-20 Gy) at a mean dose rate above 100 Gy per second, which is several orders of magnitude greater than what is currently used in routine clinical practice [92]. Even more so than the mean dose rate, the instantaneous dose rate within a radiation pulse may be a key driver of FLASH effects. Several recent studies have demonstrated that FLASH therapy induces fewer toxicities in normal tissue than conventional RT [93-95]. The mechanism of FLASH therapy is still not very clear, but several hypotheses have been proposed. Some groups have suggested that differential responses between FLASH-RT and conventional RT may be caused by the depletion of oxygen at high doses of radiation [96, 97]. In fact, multiple studies have found that FLASH therapy can deplete local oxygen and induce a short-lived protective hypoxic environment within normal healthy tissues that increases radioresistance. FLASH therapy may also modulate immune responses, which could contribute to its effect, but this needs more research [98]. FLASH-RT may directly affect immune cells or indirectly influence the tumor microenvironment. Some evidence suggests that immune cells are not preferentially spared by ultra-high dose rates similar to FLASH [99]. TGF-β is a master regulator of immune homeostasis following RT, largely mediating immunosuppressive effects . One experimental finding implies that TGF-β might be one of the key players regulating FLASH effects [95, 100]. Macrophages also play important roles in radiation-induced pulmonary fibrosis, which FLASH-RT helps to reduce [95]. FLASH-RT’s effects on immunogenic cell death remain unknown. FLASH is an exciting treatment strategy that could change the future of clinical cancer treatment. Clinical trials and future research on FLASH therapy are necessary to promote its use for curing cancer in a single fraction with less toxicity [101].
Comment #3: At the same time, I would suggest shortening the initial description of radiotherapy modalities, which can be found elsewhere in more technical contributions.
Our Response #3: We have shortened the proton therapy part and included a table highlighting the beneficial effects of proton therapy.
Comment #4: Minor comment: please provide reference to LEM model at page 4.
Our Response #4: Thank you for pointing this out; we have included a reference for the LEM model.
Reviewer 2 Report
Please see the comments below:
- There is no evidence that failure of DNA repair would be a trigger of the cytosolic DNA sensing pathway mediated immune signaling in Figure 1. Are there any mechanisms showing the fragmentation of DNA can induce immune system ?
- In Figure 2., Do abscopal effect and bystander effect generate from the same immunological effects of radiotherapy?
- In 3.3 Radiation and Cancer Vaccines, I don’t see many evidence of radiation therapy combining with cancer vaccine. Is it possible that radiation therapy combining with CAR T-cell therapy will improve cancer treatment ?
- If the cancer cells are highly radioresistance, does radiation therapy modulate RT-mediated immune response to improve the therapeutic effect in cancer treatment?
- Abscopal effect is an interesting phenomenon in radiotherapy. Are there any references or methods that can predict the enhancement of immunological response after radiotherapy?
Author Response
Our response to Reviewer 2 comments:
Comment #1: There is no evidence that failure of DNA repair would be a trigger of the cytosolic DNA sensing pathway mediated immune signaling in Figure 1. Are there any mechanisms showing the fragmentation of DNA can induce immune system?
Our Response #1: We have included the following information in our revised version:
DNA damage repair pathway factors are known to modulate immune signaling either by sensing DNA in the cytoplasm, promoting micronuclei accumulation or by releasing fragmented self-DNA to the cytoplasm [80, 81]. For example, it has been shown that nuclear-derived self-DNA accumulate in the cytoplasm in the absence of RAD51 and that triggers STING-mediated innate immune signaling in response to high-LET radiation [79]. Additionally, a recent study found that NBS1, a DNA damage sensing factor, together with its binding partners regulate cGAS binding to micronuclei in response to genotoxic stress, including ionizing radiation [82]. Thus, failure to properly repair genomic DNA can generate both fragmented DNA and micronuclei, which in turn can trigger cytosolic DNA sensing pathway–mediated immune signaling (Figure 1).
Comment #2: In Figure 2., Do abscopal effect and bystander effect generate from the same immunological effects of radiotherapy?
Our Response #2: Thank you for pointing out this important and interesting point about abscopal and bystander effect of RT. We have included the following sentences in our revised review article:
In such cases, irradiated cancer cells may release signals that can influence nearby non-irradiated cells. In contrast, the abscopal effect is a long distance and systemic effect at a distant metastatic site, which is usually mediated by immune cells such as T cells. Furthermore, RT releases cytokines, which have been shown to play a vital role in the abscopal effect [200]. Bystander effects are mediated by several mechanisms, like direct gap-junction mediated cell–cell communication via ions such as calcium or nitric oxide, though immune factors like transforming growth factor- and cytokines can also be released in the extracellular compartment and trigger local immune activation [201]. Macrophages also play an important role in the bystander effect: once they are activated by radiation, they damage neighboring cells by transferring different signaling factors [202-204]. Thus, both abscopal and bystander effects are generated by radiation-mediated immunological responses. However, bystander effects are triggered by other pathways and signaling mediators.
Evidence indicates that the abscopal effect is immune-mediated and can be induced if radiation is combined with strategies that target tumor-associated immunological dysfunction [205-207]. Several preclinical studies have tested combinations of radiation with immune modulators for their ability to induce local and systemic antitumor T cell responses [208, 209]. Some of these combinations have been translated into clinical trials too. Several researchers have elucidated the immunological effects of radiation, but there are still no methods available to predict how radiation will enhance immunological responses. We need to understand why some patients respond to radiotherapy alone or with immunomodulatory drugs and exhibit abscopal effects while others fail to do so. Some studies have tried to explain these failures by associating them with the host, the tumor or the treatment. Host criteria include hematological impairment at the time of combined treatment, neutrophil to lymphocyte ratio >4 [210], and the presence of a microbiome [211]. Tumor-specific causes include downregulation of the molecular machinery required for IFN-I activation, induction of immunosuppressive mediators like PDL-1 and TGF, and antigenic heterogeneity in different metastases [188, 212-214]. The dose and fractionation of RT can also influence the immune system, but the immunological effects of different regimens are unpredictable. Thus, it is important to identify effective biomarkers that can predict abscopal effects or show immunological response.
Comment #3: In 3.3 Radiation and Cancer Vaccines, I don’t see many evidences of radiation therapy combining with cancer vaccine. Is it possible that radiation therapy combining with CAR T-cell therapy will improve cancer treatment?
Our Response #3: Thank you for this key point. We have included a paragraph on CART-Cell therapy in our revised version:
Despite the vast body of research on the systemic immune response after RT, especially in combination with immunomodulatory drugs, there is a lack of data directly examining the synergy between radiation and CAR T-cell therapy. However, several preclinical studies have revealed the mechanism by which RT initiates tumor-specific immune responses, which can provide the basis for overcoming some of the challenges facing CAR T-cell therapy in solid tumors. High expression of tumor antigen as compared to the normal tissue is very important for this therapy. RT has the potential to increase tumor-associated antigens and cell surface receptors. One study reported that RT led to a dose-dependent increase in the expression of MHC1 molecules on the surface of tumor cells [167]. We have mentioned above that RT releases tumor-associated antigens like CEA, so it is interesting that CAR T cells directed against CEA are in clinical trial. Mesothelin is another target of interest in a study of CAR T-cell therapy for lung cancer. One study observed that mesothelin expression per cell was higher in the irradiated group than in the control group [168]. RT can target other barriers to CAR T-cell therapy, such as the trafficking of CAR T cells into solid tumors. RT can upregulate adhesion molecules like ICAM1 and VCAM1 to promote adhesion [169, 170]. Thus, even as the role of RT in CAR T-cell therapy is still evolving, we can say that RT would be a strong adjunct for this therapy.
Comment #4: If the cancer cells are highly radioresistance, does radiation therapy modulate RT-mediated immune response to improve the therapeutic effect in cancer treatment?
Our Response #4: We have included the following paragraph in our revised version:
To improve the efficacy of radiotherapy, it is important to understand the mechanisms of radioresistance. RT mainly affects and damages DNA, but upregulation of the DNA damage response is associated with radioresistance [102, 103]. Mitotic cells are hypersensitive to radiation because they inactivate DSB repair [104], though B16-BL6 and PANC-1 cells have shown high radiosensitivity during the early S-phase [105]. We know that some tumors are more radiosensitive than others, but little is known about the roles that the immune response plays across a range of radiosensitive and radioresistant cells. One study reported that radiation reprogrammed the tumor microenvironment in the parental tumor, but resistant tumors were not affected much. The parental tumor showed more CD8+ T cell infiltration than resistant cells. Along with that, T cell chemokines that play an important role in the immune response were also more present in parental cells than in radioresistant cells. This raises the possibility that CD8+ T cell infiltration after radiotherapy might be related to tumor radiosensitivity in vivo [106]. Thus, the radioresistance and radiosensitivity of tumors are important parameters that determine the extent to which radiation activates the immune response. Radiation initiates immune responses by fragmented DNA and tumor cell death. Cancer cells that are resistant to radiation will have less of an immune response than radiosensitive cells.
Comment #5: Abscopal effect is an interesting phenomenon in radiotherapy. Are there any references or methods that can predict the enhancement of immunological response after radiotherapy?
Our Response #5: Yes, we agree with the reviewer that it is very important to predict enhancement of immunological response after radiotherapy. That would be quite helpful in clinical set up. We have included two paragraphs on abscopal effects. Please refer our response #2 above.
Round 2
Reviewer 2 Report
The authors have answered my comments. Thank you.